# Pretraining with Random Noise for Fast and Robust Learning without Weight Transport

**Jeonghwan Cheon**[1]  **Sang Wan Lee**[1,2,3]  **Se-Bum Paik**[1]

[1]Department of Brain and Cognitive Sciences,
[2]Graduate School of Data Science, [3]Kim Jaechul Graduate School of AI
Korea Advanced Institute of Science and Technology, Daejeon, Republic of Korea
{jeonghwan518, sangwan, sbpaik}@kaist.ac.kr

## Abstract

The brain prepares for learning even before interacting with the environment, by refining and optimizing its structures through spontaneous neural activity that resembles random noise. However, the mechanism of such a process has yet to be understood, and it is unclear whether this process can benefit the algorithm of machine learning. Here, we study this issue using a neural network with a feedback alignment algorithm, demonstrating that pretraining neural networks with random noise increases the learning efficiency as well as generalization abilities without weight transport. First, we found that random noise training modifies forward weights to match backward synaptic feedback, which is necessary for teaching errors by feedback alignment. As a result, a network with pre-aligned weights learns notably faster and reaches higher accuracy than a network without random noise training, even comparable to the backpropagation algorithm. We also found that the effective dimensionality of weights decreases in a network pretrained with random noise. This pre-regularization allows the network to learn simple solutions of a low rank, reducing the generalization error during subsequent training. This also enables the network to robustly generalize a novel, out-of-distribution dataset. Lastly, we confirmed that random noise pretraining reduces the amount of meta-loss, enhancing the network ability to adapt to various tasks. Overall, our results suggest that random noise training with feedback alignment offers a straightforward yet effective method of pretraining that facilitates quick and reliable learning without weight transport.

## 1 Introduction

The brain refines its network structure and synaptic connections even before birth, without exposure to sensory stimuli [1–4]. In the early developmental stages, the spontaneous neuronal activity that appears in various brain regions is considered to play a critical role during the development of neuronal circuits by pruning neural wiring and adjusting synaptic plasticity [5–8]. If this activity is disrupted during the developmental stages, the outcome can be long-lasting neuronal deficits [9–11]. Computational studies suggest that such a refined network structure enables certain crucial functions of the brain, such as initializing function [12–15] and structure [16, 17]. These experimental and theoretical studies commonly indicate that spontaneous, random neuronal activity plays a critical role in the development of the biological neural network before data are encountered by the network. However, the detailed mechanism of how these prenatal processes contribute to learning after birth, i.e., with subsequent sensory stimuli, remains elusive.

At the synaptic level, learning can be defined as the process by which the brain adjusts the strength of synaptic connections between neurons to optimize the network for a specific task [18–20]. The synaptic weights of each neuron can change to minimize the error between the expected and the

38th Conference on Neural Information Processing Systems (NeurIPS 2024).

actual output of a task, often referred to as the credit assignment problem [21–23]. However, in general, it is not well known how individual neurons modify these synaptic connections and thus achieve a network goal under a condition in which numerous neurons are linked in multiple layers. In other words, how neurons can estimate errors to modify their synaptic connections during learning remains unknown.

In machine learning, backpropagation algorithms have successfully addressed this issue – even in deep neural networks [24–26]. Backpropagation can provide feedback with regard to forward errors through the symmetric copying of forward weights via a backward process. During this process, a structural constraint, i.e., symmetric forward and backward weights, is necessary to assign proper error values to individual neurons [27–29]. However, this process appears to be biologically implausible due to the weight transport problem [23, 30–32], in which individual neurons must somehow be aware of the exact synaptic connections of their downstream layers to update their weights, a state considered to be practically impossible in a biological brain.

An alternative algorithm, feedback alignment, achieves successful network training even without weight transport by employing fixed random feedback pathways [32]. This study shows that a network can align its weights to synaptic feedback during data training, and this simple process enables error backpropagation. It has been shown that soft alignment between forward and backward weights, which can be achieved during learning data, is enough to back-propagate errors. This finding may provide a biologically plausible scenario in which the credit assignment problem can be resolved, yet there is an issue remaining — the process requires massive data learning to develop the structural constraint, and it significantly underperforms compared to backpropagation on challenging tasks [33, 34]. This cannot be addressed even with currently known advanced learning rules [35–40].

This situation is contradictory to the notion that the brain can learn even with very limited experience in the initial stages of life [41–44]. Thus, the question arises as to how early brains can estimate and assign errors for learning with limited experience. To address this issue, here we focus on the role of spontaneous activity at the prenatal stage in the brain, showing that training random noise, which mimics spontaneous random activity in prenatal brains, is a possible solution; random noise training aligns the forward weights to synaptic feedback, enabling precise credit assignment and fast learning. We also observed that random noise training can pre-regulate the weights and enable robust generalization. Our findings suggest that random noise training may be a core mechanism of prenatal learning in biological brains and that it may provide a simple algorithm for the preconditioning of artificial neural networks for fast and robust learning without the weight transport process.

## 2 Preliminaries

Biological and artificial neural networks have different structures and functionalities, but they share certain factors in common, such that information is processed through hierarchical layers of neurons with a nonlinear response function. In the current study, we consider a multi-layer feedforward neural network for pattern classification, $f_\theta : \mathbb{R}^m \to \mathbb{R}^d$, parameterized by $\theta = \{\mathbf{W}_l, \mathbf{b}_l\}_{l=0}^{L-1}$. It takes input $\mathbf{x} \in \mathbb{R}^m$ and outputs a vector $\mathbf{y} \in \mathbb{R}^d$ with $L$ layers. Through a forward pass, the network computes a hidden layer output by propagating the input through the network layers, as follows:

$$\mathbf{o}_{l+1} = \mathbf{W}_l \mathbf{h}_l + \mathbf{b}_l, \quad \mathbf{h}_{l+1} = \phi(\mathbf{o}_{l+1}) \tag{1}$$

, where $\mathbf{W}_l$ is the forward weights, $\mathbf{b}_l$ is the bias vector, and $\phi$ is the nonlinear activation function. In the first layer $l = 0$, $\mathbf{h}_l = \mathbf{x}$. We used a rectified linear unit (ReLU) activation function, $\phi(x) = \max(0, x)$. In the last layer $l = L - 1$, we used a softmax function, $\phi_y(x) = \text{softmax}(x) = \{e^{x_i} / \sum_{j=1}^d e^{x_j}\}_{i=1}^d$. Thus, the network outputs a probability distribution over $d$ classes. After the forward pass, the amount of error is calculated by measuring the difference between the network output $f_\theta(\mathbf{x})$ and the target label $\mathbf{y}$. We used the cross-entropy loss [45], which is defined as follows:

$$\mathcal{L}(\theta) = -\frac{1}{N} \sum_{i=1}^N \sum_{j=1}^d \mathbf{y}_{ij} \log f_\theta(\mathbf{x}_i)_j \tag{2}$$

, where $N$ is the number of samples, $d$ is the number of classes, and $\mathbf{y}_{ij}$ is the target label for the $i$-th sample and the $j$-th class. The purpose of learning is to minimize the error $\mathcal{L}(\theta)$. To achieve this, the network parameters $\theta$ are adjusted by assigning credit to the weights that contribute to the error, which is known as the credit assignment problem.

## 2.1 Backpropagation and weight transport problem

To solve the credit assignment problem, backpropagation (BP) [24] computes the gradient of errors with respect to the weights and uses it as a teaching signal to modulate the aforementioned parameters. The gradient is calculated by the chain rule, with propagation from the output layer to the input layer, as follows:

$$\delta_L = \frac{\partial \mathcal{L}}{\partial \mathbf{o}_L} = f_\theta(\mathbf{x}) - \mathbf{y}, \quad \delta_l = \frac{\partial \mathcal{L}}{\partial \mathbf{o}_l} = (\mathbf{W}_l^T \delta_{l+1}) \odot \phi'(\mathbf{o}_l) \tag{3}$$

, where $\delta_l$ is the error signal at layer $l$, $\phi'$ is the derivative of the activation function, and $\odot$ denotes the element-wise product. The weight update rule is given by

$$\Delta \mathbf{W}_l = -\eta \delta_{l+1} \mathbf{h}_l^T \tag{4}$$

, where $\eta$ is the learning rate. The backpropagation algorithm successfully solves the credit assignment problem, but it requires heavy computation to use the complete information of the synaptic weights of the next layer to update the current weights. Notably, backpropagation is considered as biologically implausible, because it is impossible, in the brain, to transmit the synaptic weights from the next layer to the current layer. This is known as the weight transport problem [30, 31].

## 2.2 Feedback alignment

To address the weight transport problem, the idea of feedback alignment (FA) [32] was proposed as a biologically plausible alternative to backpropagation. In feedback alignment, the backward synaptic feedback is replaced with a random, fixed weight matrix $\mathbf{B}_l$ in the feedback path, as follows:

$$\delta_l = \frac{\partial \mathcal{L}}{\partial \mathbf{o}_l} = (\mathbf{B}_l \delta_{l+1}) \odot \phi'(\mathbf{o}_l). \tag{5}$$

The only difference between backpropagation and feedback alignment is the replacement of the transpose of the forward weights $\mathbf{W}_l$ with the fixed random feedback weights $\mathbf{B}_l$ to calculate the error signal. The fact that the network can learn tasks from error teaching signals that are calculated from random feedback is explained by the observation that the network modifies the forward weights $\mathbf{W}_l$ to match the transpose of the feedback weights $\mathbf{B}_l$ roughly during training. This makes the error teaching signal (5) similar to backpropagation (3), thus enabling the network to learn the task.

## 3 Random noise pretraining with feedback alignment

---
**Algorithm 1** Random noise pretraining

---
1: **procedure** RANDOM NOISE PRETRAINING($network\ f_\theta : \mathbb{R}^m \to \mathbb{R}^d$)
2:     **for each** epoch **do**
3:         **for each** batch **do**
4:             $\mathbf{x} \sim \mathcal{N}(\mu, \sigma^2),\ \mathbf{y} \sim \mathcal{U}(0, d-1)$            ▷ sampling random noise
5:             $\mathcal{L}(\theta) = \text{Loss}(f_\theta(\mathbf{x}), \mathbf{y})$         ▷ forward pass, equation (1), (2)
6:             $\delta_L = \frac{\partial \mathcal{L}}{\partial \mathbf{o}_L}$         ▷ compute error
7:             **for** layer $l = L$-1 **to** 0 **do**
8:                 $\mathbf{W}_l = \mathbf{W}_l - \eta\,\delta_{l+1}\,\mathbf{h}_l^T$         ▷ update weights, equation (4)
9:                 $\boldsymbol{\delta}_l = (\mathbf{B}_l\,\boldsymbol{\delta}_{l+1}) \odot \phi'(\mathbf{o}_l)$         ▷ compute error, equation (5)

---

During the developmental stage, spontaneous neural activity in the brain plays a critical role in shaping and refining neural circuits. Initially wired immature neural circuits undergo modifications of their connections through the processes of regulated cell formation, apoptosis, and synapse refinement through spontaneous neural activity [4–8]. These pre-sensory activities and development processes are universal across sensory modalities, such as the visual [46, 47], auditory [48, 49], and sensorimotor systems [50, 51]. We focus here on a few characteristics of spontaneous neural activity in the brain. Spontaneous neural activity is not correlated to external stimuli but can refine and optimize neural circuits, before interaction with the external world can take place.

Here, we propose a type of random training that is inspired by the spontaneous and prenatal neural activity in the brain to pretrain the neural network (Algorithm 1). In every iteration, we sampled

random noise inputs $\mathbf{x}$ from a Gaussian distribution $\mathcal{N}(0, I)$ and random labels $\mathbf{y}$ from a discrete uniform distribution $\mathcal{U}(0, N_{\mathrm{readout}} - 1)$, without any correlation. The network $f_\theta$ was initialized with random weights and trained with the feedback alignment algorithm. In this study, we examined the effects of random noise training on the subsequent learning processes in model neural networks to understand the potential benefits of pretraining with random noise in biological brains and whether this strategy is applicable to machine learning algorithms.

## 4 Results

### 4.1 Weight alignment to synaptic feedback during random noise training

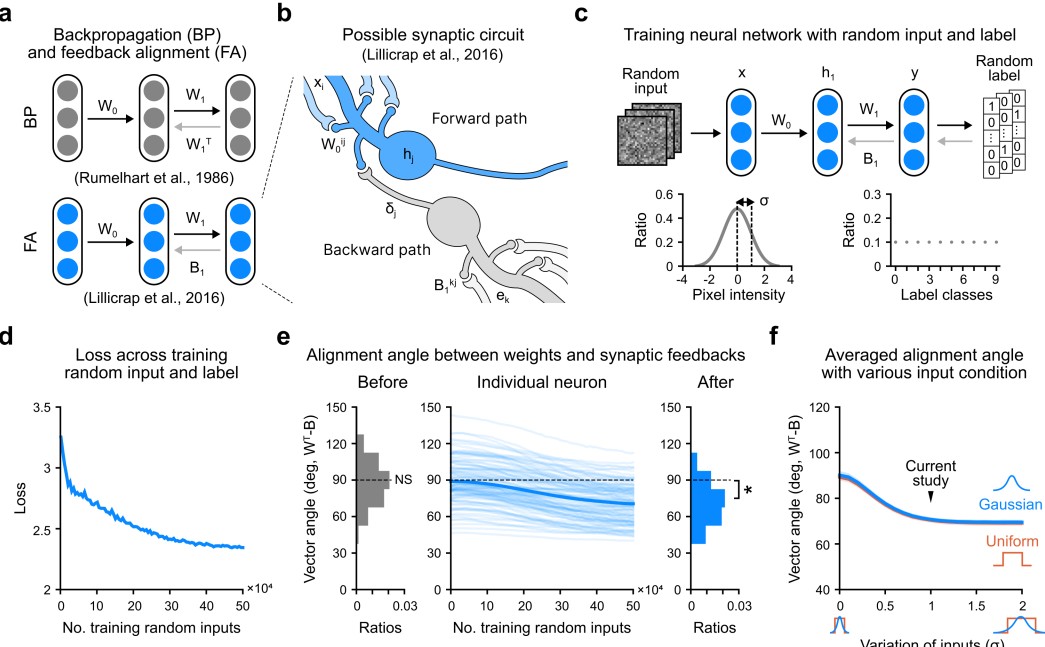

Figure 1: Weight alignment to randomly fixed synaptic feedback induced through random noise training. (a) Forward and backward pathways of backpropagation and feedback alignment. (b) Possible scenario of the feedback alignment algorithm in a biological synaptic circuit. (c) Schematic of random training, where the input $\mathbf{x}$ and label $\mathbf{y}$ are randomly sampled and paired in each iteration. (d) Cross-entropy loss during random training. (e) Alignment angle between forward weights and synaptic feedbacks in the last layer. (f) Alignment angle with various random input conditions.

To simulate a neural network initially wired by random weights and fixed random synaptic feedback, we adopted a network setting from the feedback alignment algorithm (Figure 1a) in which the weight transport problem can be avoided through the use of fixed random synaptic feedback. Thus, unlike backpropagation, this process is considered possible to exist in biological neural networks with local synaptic connections (Figure 1b). We used a two-layer feedforward neural network with ReLU nonlinearity for classification, $f_\theta : \mathbb{R}^{784} \to \mathbb{R}^{10}$ with 100 neurons in the hidden layer. By means of random noise training (Algorithm 1), we trained the neural network with random inputs sampled from a Gaussian distribution $\mathcal{N}(0, I)$, with labels also randomly sampled independently (Figure 1c).

We observed that the training loss decreased noticeably during random training, even in the absence of meaningful data and even when $\mathbf{x}$ and $\mathbf{y}$ are randomly paired (Figure 1d). During the random training process, we focused on the alignment between the forward weights and the synaptic feedback. As described in the literature [32], the alignment of $\mathbf{W}_l$ and $\mathbf{B}_l$, i.e., similarity between $\delta_{\mathrm{BP}}$ and $\delta_{\mathrm{FA}}$, is crucial for calculating the error teaching signal precisely. To evaluate the alignment, we used cosine similarity, which is widely used for measuring the distance between two vectors.

**Definition.** Given the forward weights $\mathbf{W}_l \in \mathbb{R}^{m \times n}$ and backward weights $\mathbf{B}_l \in \mathbb{R}^{n \times m}$, we measured alignment using cosine similarity. We define that $\mathbf{W}_l$ and $\mathbf{B}_l$ as aligned if the angle $\angle(\mathbf{W}_l^T)_i, (\mathbf{B}_l)_i$ is significantly smaller than 90 degrees.

Notably, we found that the weights of neurons are aligned to the corresponding synaptic feedback weights during the random training process (Figure 1e). We also observed that the angle between the forward weights and synaptic feedback of individual neurons in the hidden layer decreased asymptotically during random training. In a randomly initialized network, the alignment angle appeared to be close to 90°, demonstrating that the backward error signal is randomly distributed (Figure 1e, left, alignment angle in an untrained network vs. 90°, $n = 100$, one-sample t-test, NS, $P = 0.492$). However, after random training, the alignment angle decreased significantly, implying that the backward teaching signal becomes valid to back-propagate errors (Figure 1e, right, alignment angle in an untrained network vs. a randomly trained network, $n = 100$, two-sample t-test, $^*P < 0.001$). We confirmed that this is not simply due to input bias under a particular condition but is reproduced robustly with various input conditions (Figure 1f). These results suggest that neural networks can pre-learn how to back-propagate errors through random noise training.

## 4.2 Pretraining random noise enables fast learning during subsequent data training

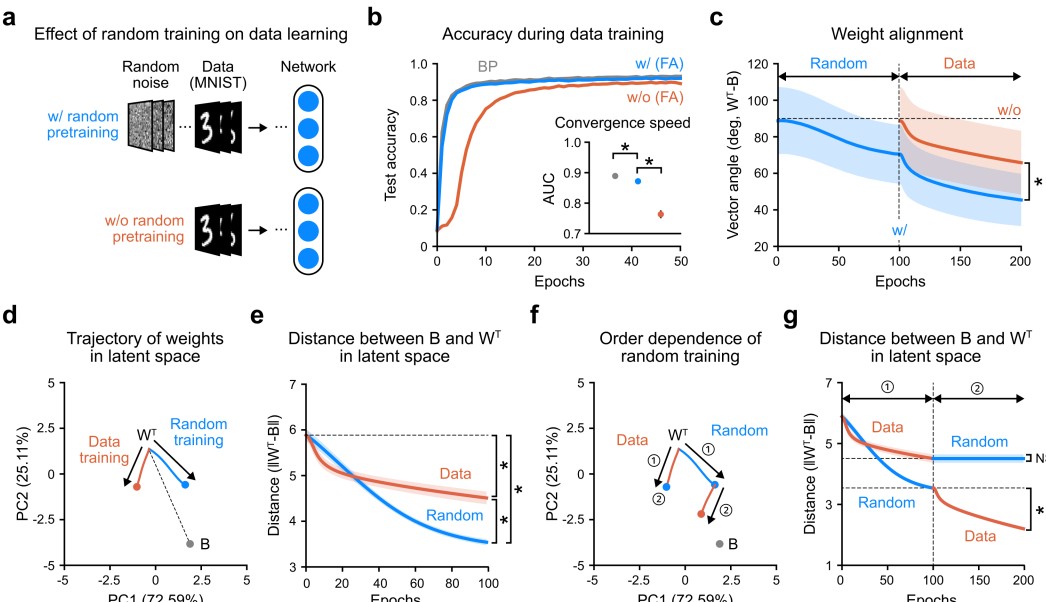

Figure 2: Effect of random noise pretraining on subsequent data training. (a) Design of the MNIST classification task to investigate the effect of random training. (b) Test accuracy during the training process, where the inset demonstrates the convergence speed of each training method, calculated by the AUC of the test accuracy. (c) Alignment angle between weights and synaptic feedback across random training and data training. (d) Trajectory of weights ($\mathbf{W}_1$) toward synaptic feedback ($\mathbf{B}_1$) in latent space obtained by PCA for random and data training. (e) Distance between the weights ($\mathbf{W}_1$) and the synaptic feedback ($\mathbf{B}_1$). (f) Order dependence of the trajectory of the weights ($\mathbf{W}_1$). (g) Distance between the weights ($\mathbf{W}_1$) and the synaptic feedback ($\mathbf{B}_1$) for different orders of random and data trainings.

Next, we compared networks with and without random pretraining in terms of subsequent data training outcomes (Figure 2a). We trained the networks using a subset of the MNIST dataset [52], a widely used benchmark for image classification. We found that a randomly pretrained network learns the data more quickly and achieves higher accuracy compared to a network that is not randomly pretrained (Figure 2b). To quantify the speed of learning, we calculated the area under the curve (AUC) of the test accuracy and found that the convergence of the randomly pretrained network is significantly faster than that in the network without random pretraining (Figure 2b, inset, w/o vs. w/ random pretraining (FA), $n_{\text{net}} = 10$, t-test, $^*P < 0.001$). Notably, the convergence speed of the randomly pretrained network appeared comparable to that of the network trained with backpropagation (Figure

2b, inset, w/ random pretraining (FA) vs. BP, $n_{net} = 10$, t-test, $^*P < 0.001$). We also observed that the weight alignment gap between untrained and randomly pretrained networks is maintained during data training (Figure 2c). As a result, at the end of the data training step, the alignment angle of the randomly trained network was significantly smaller than that of the untrained network (Figure 2c, w/o vs. w/ random pretraining, $n = 100$, t-test, $^*P < 0.001$). This result suggests that a combination of random pretraining and subsequent data training can enhance the weight alignment, which leads to more precise error teaching.

To understand the weight update dynamics by random and data training, we visualized the trajectory of weights in latent space as obtained by a principal component analysis (PCA) [53] (Figure 2d). We conducted PCA on the weights of the last layer ($\mathbf{W}_1$) for the random and data training conditions. First, we confirmed that in both random and data training, the weights become closer to synaptic feedback (Figure 2e, untrained vs. data training, $n_{net} = 10$, t-test, $^*P < 0.001$; untrained vs. random training, $n_{net} = 10$, t-test, $^*P < 0.001$; random vs. data training, $n_{net} = 10$, t-test, $^*P < 0.001$). Notably, we observed that the updated trajectory of weights by random training and data training have different directions in the principal component space and that the effects of random training depend on the order of the random and data training (Figure 2f, g) — the enhancement of weight alignment was more significant when data training was performed after random training (Figure 2g, random training vs. random and data training, $n_{net} = 10$, t-test, $^*P < 0.001$) compared to when training is done in a reversed order (Figure 2g, data training vs. data and random training, $n_{net} = 10$, t-test, NS, $P = 0.999$). Particularly, when we trained the network with data first, subsequent random training could not move the weights; thus, the weights did not become closer to synaptic feedback. This result suggests that weight alignment by random noise pretraining cannot be replaced by data training and that it is crucial to perform random training prior to data training.

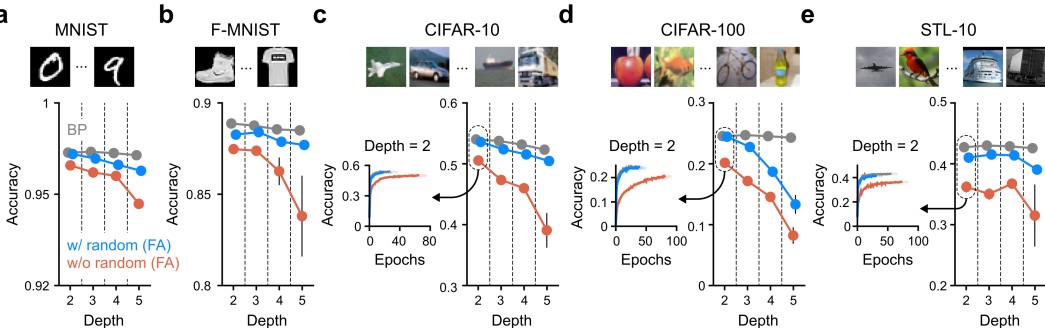

Figure 3: Comparison of model performance across different image datasets and network depths. (a-e) Final accuracy after convergence. Experiments were conducted with networks of varying depths on different tasks: (a) MNIST, (b) Fashion-MNIST, (c) CIFAR-10, (d) CIFAR-100, and (e) STL-10.

Table 1: Performance of the two-layer model for each dataset (MNIST, Fashion-MNIST, CIFAR-10, CIFAR-100, STL-10). Each performance value (%) is presented as the mean $\pm$ standard deviation from three trials. Extended results with various model depths can be found in the Appendix.

|  |  | MNIST | F-MNIST | CIFAR-10 | CIFAR-100 | STL-10 |
|---|---|---|---|---|---|---|
|  | BP | $97.82 \pm 0.03$ | $88.87 \pm 0.03$ | $54.01 \pm 0.20$ | $24.55 \pm 0.10$ | $42.72 \pm 0.20$ |
| FA | w/o | $97.26 \pm 0.07$ | $87.47 \pm 0.25$ | $50.54 \pm 0.22$ | $20.17 \pm 0.30$ | $36.21 \pm 0.91$ |
|  | w/ | $97.76 \pm 0.07$ | $88.26 \pm 0.07$ | $53.58 \pm 0.12$ | $24.45 \pm 0.10$ | $41.01 \pm 0.16$ |
|  | $\Delta$ACC | ▲$0.49 \pm 0.06$ | ▲$0.79 \pm 0.31$ | ▲$3.04 \pm 0.20$ | ▲$4.28 \pm 0.38$ | ▲$4.81 \pm 0.86$ |

Next, we further investigated the model's classification performance across various image datasets and network depths (Figure 3, Table 1). In earlier experiments with two-layer networks and MNIST, we showed that random noise pretraining enhances both the learning speed and accuracy of networks to levels comparable with backpropagation. We extended these experiments to networks of varying depths (Figure 3a) and confirmed that the benefits of random noise pretraining generalize to deeper networks. Additionally, we evaluated performance across various image datasets, including Fashion-MNIST [54] (Figure 3b), CIFAR-10 [55] (Figure 3c), CIFAR-100 [55] (Figure 3d), and STL-10

[56] (Figure 3e). We found that random noise pretraining significantly narrows the performance gap between feedback alignment and backpropagation across different datasets and depths. These results suggest that pretraining with random noise can serve as a general strategy for improving the performance of neural networks trained with feedback alignment algorithms, making it comparable to backpropagation.

## 4.3   Pre-regularization by random noise training enables robust generalization

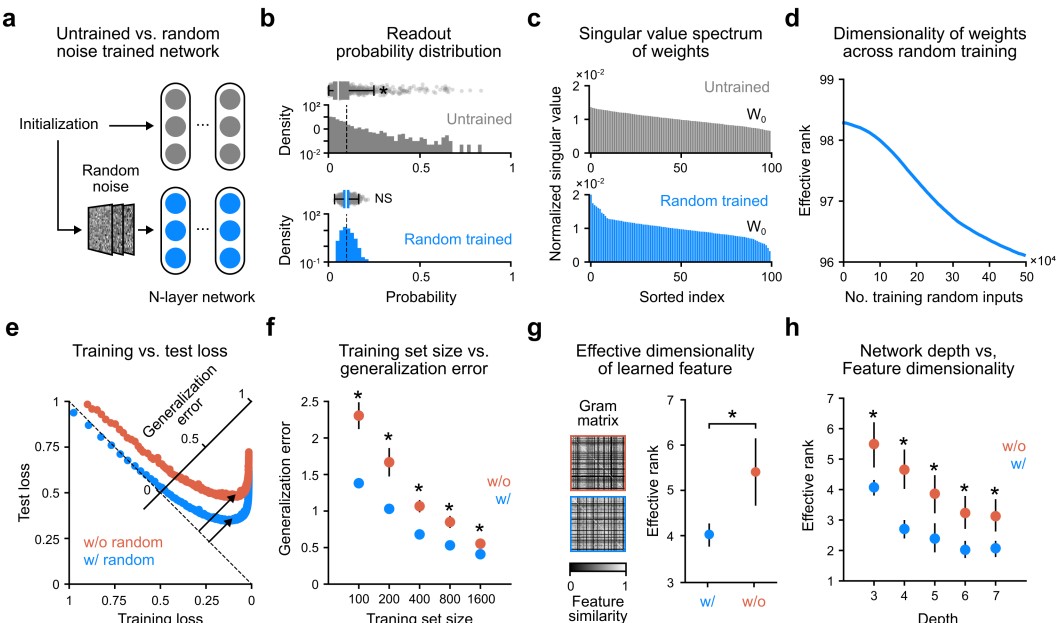

Figure 4: Pre-regularization by random noise training enhances generalization (a) Untrained network and pre-regularized network through random noise training. (b) Distribution of the readout probability. (c) Singular value spectrum of the forward weights. (d) Effective rank of forward weights during random noise training. (e) Generalization error between the training error and test error (training set size: 1600, network depth: 3). (f) Generalization error for various training set sizes (network depth: 3). (g) Effective dimensionality of the Gram matrix, the cosine similarity of feature vectors across neurons (training set size: 1600, network depth: 3). (h) Effective dimensionality of the Gram matrix for various network depths (training set size: 1600).

Next, we compared the difference between an untrained network and a randomly trained network in terms of their activation and weight (Figure 4a). First, we found that the readout probability of the untrained network is distributed over a wide range (Figure 4b, top, untrained vs. chance level, $n = 1000$, Wilcoxon signed-rank test, $*P < 0.001$), whereas that of the randomly trained network is well regularized, close to the chance level (Figure 4b, bottom, random noise trained vs. chance level, $n = 1000$, Wilcoxon signed-rank test, NS, $P = 0.096$). We also observed that the singular value spectrum of forward weights changes significantly by random training (Figure 4c) such that a small portion of singular values become dominant in the randomly trained network. To measure the effective dimensionality of the weights quantitatively, we used the effective rank of the weights.

**Definition.** Given a matrix $\mathbf{A} \in \mathbb{R}^{m \times n}$ is decomposed into $\mathbf{A} = U\Sigma V^T$ by singular value decomposition (SVD), the singular values are $\{\sigma_i\}_{i=1}^{min(m,n)}$ sorted in descending order. The effective rank $\rho$ is defined as the Shannon entropy of the normalized singular values, $\rho = -\sum_i \bar{\sigma}_i \log \bar{\sigma}_i$, where $\bar{\sigma}_i = \sigma_i / \sum_i \sigma_i$. Without loss of generality, we used the effective rank as the exponential of $\rho$ [57].

We observed that the effective rank of forward weights decreased significantly during random training (Figure 4d), implying that random training regularizes the weights initially and predisposes the network to learn simple solutions of a low rank. Given the notion that low-rank solutions show better generalization performance outcomes, we hypothesized that this pre-regularization by random training enables robust generalization during subsequent data training by inducing low-rank solutions.

To test the generalization ability of the network, we measured the gap between the training error and the test error during subsequent data training. We observed that the generalization error was noticeably lower in a randomly pretrained network compared to a network trained solely on the data (Figure 4e) and that this tendency was maintained with variations of the training set size (Figure 4f, w/o vs. w/ random pretraining, $n_{net} = 10$, t-test, $^*P < 0.001$). This result suggests that pre-regularization by random pretraining can enable robust generalization during subsequent data training.

Next, we compared the representation of learned features in a randomly pretrained network and a network trained without random pretraining. We used the Gram matrix, defined as the cosine similarity of feature vectors across neurons. Notably, we found that the effective rank of the Gram matrix was significantly lower in a randomly pretrained network compared to an untrained network after subsequent data training (Figure 4g, w/o vs. w/ random pretraining, $n_{net} = 10$, t-test, $^*P < 0.001$) and that this tendency was maintained regardless of the network depth (Figure 4h, w/o vs. w/ random pretraining, $n_{net} = 10$, t-test, $^*P < 0.001$). This finding suggests that pre-regularization by random training can enable networks to learn simpler solutions, leading to better generalization performance during subsequent data training.

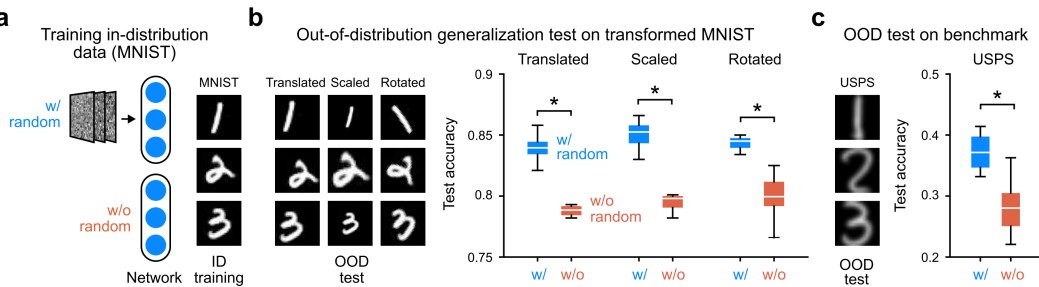

Figure 5: Robust generalization of "out-of-distribution" tasks in randomly pretrained networks. (a) Training in-distribution data (MNIST) in untrained and randomly pretrained networks. (b) Out-of-distribution generalization tests on transformed MNIST. (c) Out-of-distribution generalization tests on USPS dataset.

We also tested the generalization performance of the networks for "out-of-distribution" tasks by training the network with the MNIST dataset and testing it with various out-of-distribution tasks (Figure 5a). First, we generated a MNIST dataset of translated, rotated, and scaled images and then used these images as out-of-distribution tasks (Figure 5b, left). We observed that a randomly pretrained network showed significantly higher test accuracy on out-of-distribution tasks than a network trained without random pretraining (Figure 5b, right, w/o vs. w/ random pretraining, $n_{net} = 10$, t-test, $^*P < 0.001$). We also observed that the randomly pretrained network showed higher test accuracy on the USPS dataset [58], which is a widely used benchmark dataset for out-of-distribution tasks (Figure 5c, w/o vs. w/ random pretraining, $n_{net} = 10$, t-test, $^*P < 0.001$). This result suggests that pre-regularization by random pretraining enables robust out-of-distribution generalization during subsequent data training.

### 4.4 Task-agnostic fast learning for various tasks by a network pretrained with random noise

Lastly, we examined whether random training is generally beneficial for various tasks. We compared the task adaptation capacity of an untrained network and a randomly pretrained network on three tasks: (1) MNIST classification [52], (2) Fashion-MNIST [54], (3) Kuzushiji-MNIST [59] (Figure 6a, left). To measure the ability of fast adaptation to various tasks quantitatively, we computed the meta-loss, as suggested in a previous study of meta-learning.

**Definition.** Given the task distribution $\mathcal{T} \in \{\mathcal{T}_i\}_{i=1}^{n}$, the meta-loss of network $f_\theta$ is defined as $\mathcal{L}_{meta}(\theta) = \sum_{\mathcal{T}_i \in \mathcal{T}} \mathcal{L}_{\mathcal{T}_i}(\theta_i')$, where $\mathcal{L}_{\mathcal{T}_i}(\theta_i')$ denotes the loss of the task $\mathcal{T}_i$ and $\theta_i'$ is the adapted parameter for $\mathcal{T}_i$ [60].

We observed that the meta-loss decreased gradually during the random training process (Figure 6a, right). Considering that the training was solely performed with random inputs and labels on the three tasks to measure the meta-loss, this result suggests that networks can learn how to adapt to various tasks without any task-specific data. Next, we trained the untrained networks and randomly pretrained

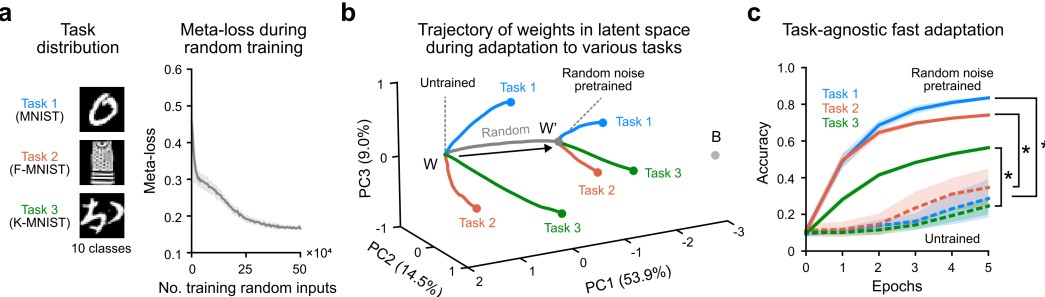

Figure 6: Task-agnostic fast learning for various tasks in randomly pretrained networks. (a) Three tasks used to test the task-agnostic property of random training, showing the meta-loss during the random training process. The meta-loss is calculated from the sum of the losses measured during adaptation to each task. (b) Trajectory of weights in the latent space for adaptation to each task of an untrained network and a randomly pretrained network. (c) Adaptation to each task of an untrained network and a randomly pretrained network.

networks on each task separately. We conducted PCA on the weights of the last layer ($\mathbf{W}_2$) to visualize the trajectory of weights in latent space during the adaptation to each task. We observed that the trajectory of weights during random noise training moves closer to synaptic feedback ($\mathbf{B}_2$), which makes the adaptation to each task more efficient (Figure 6b). This suggests that random training is task-agnostic but provides efficient and fast learning in subsequent learning. Lastly, we compared the adaptation to each task in an untrained network and a randomly pretrained network. We observed that the randomly trained network showed remarkably fast adaptation to each task compared to the untrained network (Figure 6c, w/o vs. w/ random pretraining, $n_{\text{net}} = 10$, t-test, $^*P < 0.001$). These results highlight the task-agnostic property of random training, which enables networks to quickly adapt to various tasks.

## 5    Discussion

We showed that random noise pretraining enables neural networks to learn quickly and robustly without weight transport. This finding bridges the gap between a biologically plausible learning mechanism and the conventional backpropagation algorithm, as the symmetry of forward and backward weights can easily be achieved by random noise pretraining. Moreover, the results here provide new insight into the advantage of random training as a means of preconditioning a network for robust generalization.

**Error-backpropagation without weight transport.** Early work in neuroscience identified basic learning rules, such as Hebbian learning [61] and spike-timing-dependent plasticity (STDP) [62–64]. Although these rules have been experimentally observed and are thus biologically plausible, they are not sufficient to explain the brain's remarkable learning ability thoroughly [65, 66]. On the other hand, while the backpropagation algorithms used in artificial neural networks have shown impressive learning capabilities, they are considered biologically implausible due to the weight transport problem [23]. Our results provide a new perspective on this issue, bridging the gap between these training rules. We showed that symmetry among forward and backward weights, which is necessary to back-propagate errors, can be readily developed by learning random noise, similar to that during the brain's prenatal stage. Our findings suggest a probable scenario for significantly narrowing the performance gap between previously suggested biologically plausible learning rules and backpropagation.

Recent studies explored noise as a biologically plausible mechanism to enhance learning efficiency without the need for weight transport. For instance, the weight mirror algorithm [36] uses noisy firing to align feedback weights with forward weights. Similarly, phaseless alignment learning [40] leverages layer-wise noise as an additional information carrier to achieve weight alignment. While these approaches have been reported to outperform traditional feedback alignment algorithm, merely incorporating noise into existing feedback alignment algorithms yields no improvement in learning performance [40]. In contrast, our results indicate that exposure to random noise "before encountering real data" significantly enhances vanilla feedback alignment. Unlike previous approaches, our

strategy utilizes random noise for pre-conditioning the network, preparing it to learn more effectively. This aligns with biological observations that neural noise predominates in the early stage of brain development [67], even prior to exposure to external stimuli. It is important to note that our proposed method is not limited to the feedback alignment algorithm; pretraining with random noise could be beneficial for other algorithms, which we intend to explore in our follow-up studies.

**Pre-regularization for robust generalization.** We suggest a task- and model-agnostic pretraining strategy that involves simply training the network with random noise. Notably, our results here show that random noise training can enhance the learning efficiency and generalization ability of the network, for which various tricks and techniques have been proposed to improve. We found that pretraining on random noise reduces the effective dimensionality of the weights, facilitating the learning of low-rank solutions for various tasks. Previous studies on generalization have shown that the low-rank bias of neural networks plays a crucial role in their generalization ability, a finding we have confirmed in this study [68–72]. Additionally, our results highlight that random noise pretraining functions as a form of meta-learning, enhancing the network's ability to adapt rapidly to different tasks. In contrast to previous approaches that utilized data from diverse task distributions [60], our method achieves similar effects by merely training with noise. It is important to note that the straightforward strategy of random noise training can significantly influence the network's learning dynamics - effects that previous machine learning techniques have sought to achieve. This approach may reflect a potential strategy employed by the brain to attain notable generalization capabilities. Furthermore, it suggests a novel pretraining strategy for artificial neural networks.

**Insights into developmental neuroscience.** Unlike artificial neural networks, the brain is ready to learn before encountering data. In the early developmental stage before eye-opening, spontaneous random activity emerges in the brain, which is considered essential for the normal development of early circuits [1–4]. However, the functional advantage of learning from random noise before external sensory inputs remains unclear. Our study provides a plausible scenario that the brain utilizes spontaneous random activity to pre-align the synaptic weights for error learning and pre-regularization of synaptic connections for robust generalization. Specifically, we showed that random training reduces the effective dimensionality of the weights, which can be considered as a form of pruning, as previous neuroscience studies reported that the brain's synaptic connections are pruned substantially during development, particularly dependent on spontaneous activity [4–8]. Despite the fact that the present study is based on model neural networks, the results here are consistent with a range of experimental findings in developmental neuroscience.

## 6 Broader impacts and limitations

**Broader impacts.** Feedback alignment algorithm and its advanced modifications without weight transport are motivated by the need to suggest a learning method that is compatible with deep neural networks with biological plausibility. It can be useful particularly when implemented in physical circuits, as nowadays deep learning without weight transport is utilized in neuromorphic chip engineering. Given that backpropagation requires dynamic access to memory due to weight transport, it is not free from the issue of energy inefficiency. Our results are not solely limited to demonstrating the role of biological prenatal learning but can also be extended for more practical purposes; for instance, it is a promising strategy for the preconditioning of neuromorphic chips.

**Limitations.** Although our study offers a new perspective on pretraining neural networks with random noise, some limitations must be considered. The current study focuses on results using feedforward neural networks with feedback alignment algorithms. Regarding the scalability of the method, further investigation is needed for other architectures, such as convolutional neural networks. Notably, we achieved meaningful results showing that pretraining with random noise can also benefit standard backpropagation, which will be further explored in our follow-up studies, which will emphasize random noise pretraining as a general strategy for neural network training.

## 7 Code availability

Python 3.11 (Python software foundation) with PyTorch 2.1 was used to perform the simulation. The code used in this work is available at `https://github.com/cogilab/Random`.

## Acknowledgements

This work was supported by the National Research Foundation of Korea (NRF-2022R1A2C3008991 to S.P.), the Singularity Professor Research Project of KAIST (to S.P.), the Institute of Information & communications Technology Planning & Evaluation (IITP) (RS-2023-00233251, RS-2019-II190075 to S.W.L.) and the Ministry of Science & ICT(RS-2024-00436680 to S.W.L.).

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
