# OpenReview forum: "Pretraining with Random Noise for Fast and Robust Learning without Weight Transport"
_NeurIPS.cc/2024/Conference — NeurIPS 2024 poster_

### Official Review · Reviewer_o5xh · 2024-06-28

**Soundness:** 3
**Presentation:** 3
**Contribution:** 3
**Rating:** 7
**Confidence:** 4

**Summary:**

The authors argue that networks trained with feedback alignment can be pre-trained with random input-output pairs. They demonstrate that this allows for faster learning (after the input-output pairs). They also show that the effective dimensionality of the network activity decreases if the pretraining is used, which could explain an improved generalization.

**Strengths:**

The paper provides a simple way to accelerate training with feedback alignment. The results are convincing in terms of faster learning, and I haven't seen a paper using random pre-activations. I think it's a good insight, and it is well presented.

**Weaknesses:**

There is a critical missing point in my opinion.
One is that the authors do not train until convergence, neither for the random pre-training nor for the task related data. I could think that there could be some limitations if the networks are trained for a long time:
- For example, it could be that having a too-low rank during pre-training is bad for learning later on; since the rank is related to the expresiveness of the network this would be plausible (although it might not be the case).
- It could also be that at the end of training a network trained only on data performs better at the cost of more computation (I personally don't think so, but it should be verified). Although I think I can see a performance plateau in Fig. 2 b, it should be put in logscale to see if the performance really saturated or if it can keep increasing. If it is the second, then the authors should see when does it stop increasing and whether random+data really achieves better end-of-training results.
- A related point: I haven't seen the performances in numbers. While I would not care too much for them on the main text, they should be added somewhere just to clarify that the training is done to a reasonable level; if the performance is only up to 90% or something that can be achieved with a logistic regression on MNIST then there would be a problem.
Note that this is not a critical flaw; it's still a good idea to pre-train for faster learning, even if that could potentially lead to slightly lower performances later on (at least where biology is concerned, 92% accuracy or 93% is not really a big deal).

There are also minor issues to correct or comment:
- Line 141: The loss decreased noticeably during random training. Should clarify that they mean the training loss (sorry if it is a bit pedantic)
- Definition 1has a wording problem: the claim is that Wl and Bl are aligned if the expected angle decreases asymptotically during learning. Lets say that we have W^* after learning. If I just happen to have initialized the weights as W* by chance from the beginning, the statement in lines 148 and 149 would tell me that those weights are not aligned (even though they are the same).

A few related works:
-  Error backpropagation without weight transport: While it is true that the weight transport is thought to be impossible exaclty it could be achieved by some form of hebbian-like learning, see for example Amit, Y. "Deep learning with asymmetric connections and Hebbian updates." Frontiers in computational neuroscience 13 (2019).
- STDP and backpropagation: Although it is true that STDP is not enough to explain learning, there are simple architectures that can implement learning very similar to Backprop using a rate-based version of STDP (Aceituno et al. "Learning cortical hierarchies with temporal Hebbian updates"). Importantly, that not all the architectures require weight symmetry for training (the paper menitoned has a table listing them)

Finally, I wanted to add a suggestion that I think would improve the paper. But it involves math and I don't know if the authors are up for that, so feel free to ignore it.
I think it should be possible to show that the pretraining would reduce the effective rank. In a "batch" of pre-training with S samples and C classes, backprop would push the activity to become aligned with one of the C output directions. More analytically,:
-  the weight updates are Dw = x_pre delta_post, and delta_post is always one of S random vectors.
- But there are only C classes, and thus the deltas within a class should be of a low rank statistically. More specifically
  + Compute the expectation and variance over the Jacobian,
  + for ReLus is the product of feed-forward weights from the current layer onwards, with a probability of 0.5 that a given neuron in the path will drop out of the derivative ).
  + The variance will decrease with increasing S, while the expectation will remain constant.
- Once you have c delta_post, you can be reasonably sure that the result of learning will map the activity of the previous layer into something covering only the span of delta_post.
  + This can be done by taking approximating the averaged BP rule as a delta rule.
I hope is not too unclear

**Questions:**

I don't get what is the task agnostic section supposed to show. If the point is that the network with pre-training is better than the network without pre-training in general, then why introduce concepts such as meta-loss or task-agnostic. It seems to me that is equally valid (and maybe better) to say simply that the random pre-training is great in all tasks (and exemplify by a few).

**Limitations:**

Yes. But
- 314-315: I don't get what the fact that the architecture that they mention does not use weight transport implies that more experimental studies are needed. to confirm their results. What does the lack of weight transport (which is probably true) have to do with more experiments? If they want to "prove" that weight transport is not present in cortex don't worry, no one thinks it's present. Anatomically the apical projections (top-down) are different than the basal ones anyway (bottom-up)

---

> ### Author Rebuttal · Authors · 2024-08-07
>
> > One is that the authors do not train until convergence, neither for the random pre-training nor for the task related data.
>
> **: Please find our response in Global Response, issue 1**
>
> We acknowledge the suggestion that comparisons should be made after ensuring full convergence during network training.
> - We conducted additional experiments on various datasets beyond MNIST. In these experiments, we ensured that sufficient convergence was achieved before concluding training (Global Response, issue 1).
> - Specifically, we ensured training accuracy reached saturation close to 1, or that validation accuracy no longer increased for 10 epochs, indicating the completion of learning.
> - We will update all network training and analyses performed in the manuscript according to these criteria. Importantly, even after ensuring convergence, all trends and qualitative differences remained consistent.
>
> > It could be that having a too-low rank during pre-training is bad for learning later on
>
> **We found that even with extensive random noise pre-training, the rank reduction was not dramatic enough to impede the learning of rich representations.**
> - After random noise training, the effective rank decreased only slightly in most cases (Supple. C.3). This introduces a bias towards learning low-rank solutions in subsequent training but does not significantly hinder representation learning.
> - Depending on the network structure (width and depth), certain layers experienced a decrease in effective rank during random noise pre-training, which was subsequently recovered during data training. This suggests that while initial rank may affect information representation, it can be compensated for in later stages of learning.
>
> In additional experiments, we proposed an effective pre-training length, also addressing concerns about excessively long pre-training periods (Please find our response in Global Response, issue 2).
>
> > It could also be that at the end of training a network trained only on data performs better at the cost of more computation
>
> **We quantitatively compared accuracy after achieving sufficient convergence and confirmed that random noise training generally results in higher accuracy regardless of network depth or dataset type** (Response Figs 1, 2, 3; Response Table 1).
>
> > I haven't seen the performances in numbers.
>
> **We provided numerical results for additional experiments** (Response Table 1). We will include these numerical results in a separate appendix in the final manuscript.
>
> > it's still a good idea to pre-train for faster learning, even if that could potentially lead to slightly lower performances later on.
>
> We appreciate your comments emphasizing the benefits of pre-training for faster learning, especially in biological systems. **Our results indicate that our method achieves both faster learning and higher performance.** Thus, our approach demonstrates that learning speed and accuracy need not be mutually exclusive but can be concurrently enhanced through random noise pre-training.
>
> > Minor issues (description for loss reduction, definition of alignment)
>
> We will clarify that the loss refers to "training loss" for random noise.
>
> We acknowledge that our current definition of “alignment” may not hold in some cases. We will revise the definition as follows:
> Given the forward weights W and backward weights B, we measured alignment using cosine similarity. We define W and B as aligned if \angle{W^T, B} is significantly smaller than 90 degrees.
>
> > A few related works:
>
> We will discuss the importance of weight symmetry for high-performance deep learning and also mention that there are learning architectures that do not require weight symmetry. Relevant papers will be cited in this discussion.
>
> > I think it should be possible to show that the pretraining would reduce the effective rank.
>
> We appreciate your theoretical insights, which bolster our empirical findings.
> - Your detailed suggestions for theoretical proofs are insightful, particularly in understanding how random noise training reduces effective rank, especially in scenarios with limited readout capabilities.
> - During this rebuttal period, we focused on conducting additional experiments to validate our results in broader contexts. Due to time constraints, we could not fully develop and present the theoretical proofs, despite your helpful guidance.
> - We understand that including these proofs would significantly strengthen our findings, and we aim to incorporate them into the final manuscript.
>
> > I don't get what is the task agnostic section supposed to show.
>
> We used "task-agnostic" to indicate that random noise training provides benefits in subsequent learning across various datasets. However, we acknowledge that this term might be unclear. We agree that "generally useful for various tasks" better conveys our intended meaning, as you suggested. We will replace the section title and descriptions accordingly in the manuscript.
>
> > I don't get what the fact that the architecture that they mention does not use weight transport implies that more experimental studies are needed.
>
> We appreciate your perspective and detailed explanation regarding the biological implausibility of weight transport.
> - Describing specific biological circuits, as you suggested, would provide more compelling evidence than simply stating that weight transport is biologically implausible. We will include this in Section 2, Preliminaries, where we discuss the weight transport issue and mention that in the brain, apical (top-down) and basal (bottom-up) projections are anatomically distinct.
> - We will remove this description from the limitations section.
>
> We extend our gratitude to Reviewer o5xh for recognizing the significance of our research. Your thoughtful review has encouraged us to conduct additional experiments during the rebuttal period, further validating the importance of our findings.

---

### Official Review · Reviewer_5zAn · 2024-07-04

**Soundness:** 3
**Presentation:** 3
**Contribution:** 2
**Rating:** 5
**Confidence:** 3

**Summary:**

This paper explores the idea that the brain uses spontaneous prenatal activity to optimize its structure. This is done by showing the benefits of pre-training on random noise for artificial neural networks that are trained with the feedback alignment method - a training strategy that is more biologically-plausible than traditional backpropagation. Through experiments involving networks with and without pre-training, it is shown that pre-training enables these networks to reach convergence speeds comparable to backpropagation and enables them to generalize more effectively to various tasks.

**Strengths:**

The explanations of both the theory and empirical results are clear and well explained, for the most part.
Interesting and thorough analyses were conducted to demonstrate that random pre-training of an artificial neural network trained with feedback alignment results in quicker convergence and more greater task generalization properties. I particularly appreciated the variety of studies investigating the trajectories of various metrics (e.g., accuracy, weight alignment, etc.) over time.

**Weaknesses:**

I am unclear as to what the Definition in lines 147-149 is trying to convey. Does “cosine angle” mean cosine similarity? If so, does \mathbb{E} signify this chosen loss function, or is it an expectation over the similarities between the forward weight matrices and random feedback matrices across individual neurons? Why is \angle only used in front of the forward weight matrices?
Throughout Figure 2, the authors show numerous analyses (i.e., b, c, e, and g) of how certain metrics change during training. However, the label of the X-axis says “No. of training inputs.” However, given that the various experiments outlined throughout the Appendix list the number of training epochs as 100, I am inclined to believe that that is what the authors meant instead. Further clarity in this matter would be appreciated.
A minor grievance, but referring to models without pre-training as “untrained” can be misleading.
While not critical to the overall message of the paper, Appendix sections C.4 and C.5 could be further strengthened by adding the training trajectories of a network trained with back-propagation for comparison.

**Questions:**

Figure 2.b demonstrates how a network that has undergone random pre-training converges more quickly than a network without pre-training. Do these plots account for the amount of “time” spent pre-training? If not, how much would considering the extent of pre-training close the gap in training speed between these two strategies?
What is the overall contribution this paper wishes to make? I am not clear as to whether this paper wishes to a) validate established neuroscience findings using artificial neural networks or b) use insights from the brain to propose a way to improve current learning strategies in deep learning. If a), I have trouble understanding how recreating findings from experimental neuroscience in an artificial neural network trained with a more biologically-plausible algorithm constitutes a novel research contribution. If b), the empirical results here indicate that models trained with feedback alignment improve as the forward weights become more similar to the fixed backward weights - a property that is established in traditional artificial neural networks via the symmetry in their forward and backward weights. Has pre-training on random inputs shown any benefits for networks trained with traditional backpropagation (e.g., faster convergence, faster task generalization, increased accuracy, etc.)? Perhaps this would be an interesting investigation that establishes a benefit of biologically-inspired learning for traditional neural networks.

Answers to these questions might persuade me to move my score.

**Limitations:**

Limitations are somewhat stated and addressed. As mentioned, all of the experiments are run on a feedforward network. It could be worthwhile to verify to what extent these results are reproducible when using a different architecture, such as a recurrent or convolutional neural network.

---

> ### Author Rebuttal · Authors · 2024-08-07
>
> > I am unclear as to what the Definition in lines 147-149 is trying to convey.
>
> We acknowledge that the definition in Lines 147-149 regarding alignment between forward and backward weights is unclear.
> - \mathbb{E} represents the expectation over the parallels between W and B across independent neurons.
> - \angle refers to the angle between W and B, not each of W and B.
>
> We will revise the definition of weight alignment as follows:
> Given the forward weights W and backward weights B, we used cosine similarity as a measure of alignment for individual neurons. We assert that W and B are aligned if \angle{W^T, B} is significantly smaller than 90 degrees.
>
> > However, the label of the X-axis says “No. of training inputs.”
>
> We understand that the term "No. of training inputs" is not commonly used in machine learning and can be misleading. In the final version of the manuscript, we will correct it to "epoch."
> - Initially, we used "No. of training inputs" due to the ambiguous definition of an epoch in random noise training.
> - Random noise training samples input x and label y in a random distribution. This is why we did not use "epoch" on the x-axis, which typically refers to traversing the entire dataset once.
> - To clarify, we will label the x-axis as "epoch" and explain in the text that in random noise training, samples in each epoch are newly sampled.
>
> > Referring to models without pre-training as “untrained” can be misleading.
>
> In our paper, we distinguish four network states:
> 1. Untrained (no training performed)
> 2. Only random noise trained
> 3. Data trained with an untrained network
> 4. Data trained with a random noise trained network
>
> - Most results cover states (3) and (4). In Figure 3a, 3b, and 3c, "untrained network" refers to a randomly initialized, untrained network (1), and "random trained" refers to a network trained only with random noise (2).
> - We will organize the terms and colors to clearly distinguish these four network states in the final version of the manuscript.
>
> > Appendix sections C.4 and C.5 ...
>
> As suggested, we will include results with backpropagation in Appendix C.4 and C.5.
> - We already have these results and have confirmed that the network with random noise pretraining learns much closer to backpropagation than to baseline feedback alignment.
> - This strengthens our argument that random noise training improves feedback alignment learning to levels comparable to backpropagation.
>
> > How much would considering the extent of pre-training close the gap in training speed between these two strategies?
>
> **: Please find our response in Global Response, Issue 2**
>
> As you pointed out, comparison of convergence speed in Figure 2b does not account for the "time" spent on random noise training and only considers subsequent data training.
> We agree that it is important to investigate whether random noise pretraining ensures fast convergence even when considering the time spent on pretraining. This factor can further emphasize the significance of our research.
> - We conducted additional analyses and measured the time to converge in real data learning after random noise pretraining with various lengths (Response Fig 4).
> - Our analyses showed that networks with random noise pretraining can learn data quickly, even when including the time spent on pretraining in most conditions. Additionally, this analysis allowed us to estimate the optimal pretraining length for resource-efficient learning.
>
> > What is the overall contribution this paper wishes to make?
>
> Our goal is to improve current strategies in deep learning by drawing insights from the brain.
> - There exists a significant accuracy gap between backpropagation and biologically plausible learning strategies without weight transport. While backpropagation is effective, it is computationally intensive, requiring dynamic memory access due to weight transport (access W in memory to compute backward updates).
> - As pointed out, the results of pretraining with random noise and aligning forward-backward weights resemble traditional backpropagation properties, but our approach achieves this without weight transport.
> - Therefore, our interest lies not in achieving comparable performance to backpropagation but in achieving it without weight transport. Hence, our results provide insights into achieving learning efficiency levels comparable to backpropagation without using weight transport.
> - It is noteworthy that feedback alignment (and random noise pretraining) does not require weight transport and relies solely on local information. In the context of energy-efficient neuromorphic chip engineering, feedback alignment is sometimes used for learning, albeit with some performance trade-offs. Our results demonstrate that such sacrifices are minimal in our model.
>
> We briefly discussed these arguments in the broader impact section, but we realize that our intentions were not explicitly stated. This will be clearly articulated in the final manuscript's discussion.
>
> On the other hand, we also discovered that pretraining with random noise benefits learning with backpropagation, although we did not include these results as they are not the main focus of the study.
> - For instance, results related to weight alignment (Figure 1) are specific to the feedback alignment algorithm. However, findings related to the low-rank bias of forward weights and its resulting better generalization (Figure 3, 4) are also observed in backpropagation.
> - Please note that we have identified additional benefits of random noise pretraining in backpropagation and are preparing this as part of a follow-up study.
>
> We sincerely appreciate Reviewer 5zAn's careful consideration in improving the organization and clarity of our manuscript. We particularly value the opportunity to articulate how our research aims to contribute to the field and to present additional results that highlight the strengths of our work. We hope that our rebuttal and additional experimental findings will be convincing.

---

> > ### Comment · Reviewer_5zAn · 2024-08-12
> >
> > I would like to thank the authors for their detailed rebuttal. The clarifications were helpful, and I would particularly like to commend the additional experiments and the quality of their visualizations. While I a) do not see a clear path forward for deploying feedback alignment for more difficult tasks (for instance, Fig.3 shows a clear drop-off in performance in comparison to backpropagation when moving beyond a 10 class paradigm) and b) think this paper could benefit from your results showing that pretraining with random noise benefits backpropagation as well, I ultimately believe that the changes and additional experiments implemented by the authors have produced a more coherent paper showcasing some interesting results. Therefore, I have increased the score to reflect this.

---

> > > ### Author Response · Authors · 2024-08-13
> > > **Acknowledgment**
> > >
> > > We deeply appreciate your careful review of our results. We believe the issues you raised are critical starting points for future research stemming from this manuscript, and we will certainly address these detailed points in our subsequent work following this brief revision period. Once again, thank you for your insightful comments. We hope you will be interested in our future research.

---

### Official Review · Reviewer_kKCk · 2024-07-05

**Soundness:** 3
**Presentation:** 3
**Contribution:** 3
**Rating:** 6
**Confidence:** 4

**Summary:**

The paper explores how pretraining neural networks with random noise can improve learning efficiency and generalization without relying on weight transport, inspired by spontaneous neural activity in developing biological brains.

Key Findings:
1. Random noise training aligns forward weights with synaptic feedback, enabling more precise error backpropagation without weight transport.
2. Networks pretrained with random noise learn subsequent tasks faster and achieve performance comparable to backpropagation.
3. Random noise pretraining acts as a form of regularization, reducing the effective dimensionality of weights and leading to more robust generalization.
4. Pretrained networks show better performance on out-of-distribution tasks and adapt more quickly to various tasks.

**Strengths:**

The paper investigates, to the best of my knowledge, a novel idea for pretraining neural networks to improve downstream feedback alignment. The paper is well written and easy to follow, experiments are nicely presented.

**Weaknesses:**

Unfortunately I think the evidence the paper presents is not enough to back up their claims. The bulk of the experiments only studied one-hidden neural network trained on MNIST. Results and claims will most likely be overestimated and will not hold in general. The paper also does not provide any theoretical insights why this initial pre-training might help.

**Questions:**

1) Please tune baselines. One can fool oneself very easily on the datasets and small networks you study. First, please grid search all hyperparameters in a reasonable range for the backdrop baseline and more crucially for the FA from scratch baseline. Try all common initialization schemes, and tune for them learning rate and batchsize. Please study carefully initialized random feedback weights.
2) Please study the following crucial baseline: Initialize the feedback weights B with W transpose. Your claim is essentially that when you do this you train faster than with B random. I am highly doubtful of this, and therefore your claims. I agree that your pretraining might align the weights but I am quite confident that this will not lead to better optimization. In the beginning of training weights (so only W in your case) change quickly and therefore even initial perfectly aligned weights will not bring help much.
3) How can you overcome the limitations of depth? What about layerwise local losses? If the result only hold for one-hidden layer networks, I am not confident to have this accepted at Neurips.

**Limitations:**

Please mention that your results only apply to 1-hidden layer neural networks.

---

> ### Author Rebuttal · Authors · 2024-08-07
>
> > The bulk of the experiments only studied one-hidden neural network trained on MNIST
>
> **: Please find our response in Global Response, Issue 1**
>
> **We expanded our experiments by increasing the depth of the network and confirmed that even with deeper architectures, the accuracy of the randomly noise-pretrained network consistently surpasses that of the baseline FA** (Response Figs 1, 3 and Response Table 1).
>
> Furthermore, beyond MNIST, we assessed the impact of random noise training on fMNIST, CIFAR10, CIFAR100, and STL10, consistently achieving higher accuracy (see Response Figs 2, 3 and Response Table 1).
>
> > Please tune baselines.
>
> **We are confident that our results were not obtained from different baselines as you expressed concerns.**
> - Please note that our results are based on hyperparameters typically chosen for effective learning in both FA and BP. We conducted and compared baseline FA (FA w/o), FA with random noise pretraining (FA w/), and BP under carefully controlled conditions.
> - We controlled hyperparameters such as batch size and learning rate, as well as initial forward weights, to isolate differences attributable to different learning algorithms. Specifically, in a single trial, we duplicated one randomly initialized network into three, starting with identical weights. These networks were then trained using the same hyperparameters but different learning methods and algorithms (FA w/o, FA w/, BP).
> - Additionally, both baseline feedback alignment and backpropagation showed significant performance in benchmark tests. We have presented these results numerically (Response Table 1), which we believe further supports that our findings were obtained under appropriate training conditions.
> - Regarding weight initialization, we primarily utilized He initialization since we employed ReLU as the non-linear function in the network. As backpropagation utilizes W for backward computation, we ensured that B in feedback alignment followed the same statistical characteristics as W.
>
> **Through further experiments, we demonstrated that varying hyperparameters (batch size, learning rate) and initialization within a reasonable range consistently reproduced our results** (Response Figure 5). Due to the 1-page PDF limit for the global response, we were unable to include it, but we confirmed that our results are reproducible when using tanh as the non-linear function.
>
> > Please study the following crucial baseline: Initialize the feedback weights B with W transpose.
>
> **As you suggested, we initialized B to match the transpose of W and conducted learning** (Response Fig 6). This approach resulted in gradients and alignment fairly similar to backpropagation during training. Thus, **we confirmed that this setup does not hinder learning and instead demonstrates efficiency comparable to backpropagation**.
>
> - During the learning process (especially at the beginning), weight alignment may slightly loosen but remains mostly valid.
> - This baseline example illustrates that achieving initially aligned forward and backward weights allows for learning comparable to backpropagation, even without enforced synchronization of backward weights during training.
> - It's a specific instance highlighting the validity of random noise training in aligning forward and backward weights in a biologically plausible manner before data-driven learning.
>
> > How can you overcome the limitations of depth?
>
> **Our additional experiments confirmed that random noise pretraining scales effectively to deeper networks and more challenging tasks, despite the limitations of baseline feedback alignment.**
>
> Indeed, previous studies on biologically plausible backpropagation without weight transport often focus on simple network structures and easy datasets [Lillicrap et al., Nat. Comm., 2016; Dellaferrera and Kreiman, ICML, 2022; Toosi and Issa, NeurIPS, 2023], and this limitation results in lower learning capacity and presents challenges in scaling up due to biological constraints that preclude weight transport.
>
> **Through our additional experiments, we have shown that random noise pretraining consistently outperforms baseline feedback alignment, even in deeper networks.** However, we also observed that the performance gap with backpropagation widens as the number of layers increases.
>
> - This challenge partly arises from the difficulty of achieving precise weight alignment in early layers using feedback alignment, without weight transport.
> - The concept of "layerwise local loss" may provide ideas to address this issue, potentially enabling us to scale our results more effectively and achieve performance comparable to backpropagation in very deep networks. Specifically, this approach involves optimizing consecutive 2-layer blocks independently within a deep network. We are actively exploring several strategies to narrow the performance gap between feedback alignment and backpropagation using this approach, which we are preparing as part of our follow-up project.
>
> We sincerely appreciate Reviewer kKCk's insightful comments. We acknowledge that in our initial work, we may not have sufficiently demonstrated the reproducibility and generalizability of our results, potentially leading to concerns about overestimation. Thanks to your feedback, we have conducted extensive additional experiments, which we believe have strengthened our arguments. We are grateful for your constructive criticism and we hope that our additional findings are convincing.

---

> > ### Comment · Reviewer_kKCk · 2024-08-08
> > **Thank you!**
> >
> > Thank you very much for the vast additional data. This looks convinving, I will neverthless check these numbers thoroughly over the next days and get back to you. Thank you

---

> > > ### Comment · Reviewer_kKCk · 2024-08-11
> > > **Thank you!**
> > >
> > > I raised my score accordingly - I am now convinced that the proposed method does indeed improve FA significantly. Thank you for the additional data

---

> > > > ### Author Response · Authors · 2024-08-12
> > > > **Acknowledgment**
> > > >
> > > > Thank you for the time and care you have put into reviewing our manuscript. We are grateful for your thoughtful and constructive feedback, and we appreciate your generous consideration.

---

### Author Rebuttal · Authors · 2024-08-07

We sincerely appreciate the reviewers' thorough evaluation of our manuscript and their constructive feedback. We have revised our manuscript based on their comments, including additional analyses and simulations. We are confident that our revisions address all concerns raised, further validating our results. Below, we provide detailed responses to each question. Thank you for your consideration.

## Issue 1: Validation of Model Performance Across Various Image Datasets and Network Depths

**We have ensured that our model's performance remains consistent under more general conditions by testing it across five distinct types of datasets and varying the network depth from 2 to 5 layers** (Response Table 1).

### **Global Response Table 1.** Performance of each model with depth variation (2 – 5 layers) for five different datasets (MNIST, FMINST, CIFAR10, CIFAR100, and STL10)
\* Each performance value (%) is represented as the mean ± standard deviation from three trials.

||Model depth||2 Layer|3 Layer|4 Layer|5 Layer|
|-|-|-|-|-|-|-|
|MNIST|BP||97.8±0.0|97.9±0.0|97.8±0.2|97.7±0.2|
||FA|w/o|97.3±0.1|97.0±0.1|96.8±0.2|95.6±0.2|
|||w/|97.8±0.1|97.6±0.1|97.3±0.2|97.0±0.0|
|||ΔACC|**▲0.5±0.1**|**▲0.6±0.1**|**▲0.5±0.3**|**▲1.5±0.2**|
|FMNIST|BP||88.9±0.0|88.8±0.1|88.6±0.0|88.5±0.1|
||FA|w/o|87.5±0.3|87.4±0.2|86.3±0.8|83.8±2.2|
|||w/|88.3±0.1|88.4±0.0|87.9±0.1|87.7±0.2|
|||ΔACC|**▲0.8±0.3**|**▲1.0±0.2**|**▲1.6±0.9**|**▲3.9±2.4**|
|CIFAR10|BP||54.0±0.2|53.8±0.2|53.2±0.2|52.3±0.0|
||FA|w/o|50.5±0.2|47.3±0.8|46.0±0.3|39.1±2.9|
|||w/|53.6±0.1|52.4±0.1|51.5±0.3|50.5±0.8|
|||ΔACC|**▲3.0±0.2**|**▲5.0±0.8**|**▲5.6±0.1**|**▲11.4±2.5**|
|CIFAR100|BP||24.6±0.1|24.7±0.0|24.5±0.1|24.3±0.1|
||FA|w/o|20.2±0.3|17.2±0.5|14.6±0.1|8.2±1.4|
|||w/|24.5±0.1|22.8±0.4|18.7±0.7|13.3±1.5|
|||ΔACC|**▲4.3±0.4**|**▲5.6±0.2**|**▲4.1±0.5**|**▲5.1±1.3**|
|STL10|BP||42.7±0.2|43.0±0.3|42.9±0.1|42.6±0.2|
||FA|w/o|36.2±0.9|35.0±0.9|36.7±0.2|31.5±5.1|
|||w/|41.0±0.2|41.5±0.2|41.4±0.1|39.0±0.6|
|||ΔACC|**▲4.8±0.9**|**▲6.5±1.0**|**▲4.6±0.4**|**▲7.5±5.4**|

We benchmarked the final accuracy of networks trained using baseline feedback alignment (FA, w/o), feedback alignment with random noise pretraining (FA, w/: our model), and backpropagation (BP) (Response Table 1). Specifically, **we evaluated these models at depths ranging from 2 to 5 layers (Response Fig. 1). Furthermore, we compared these models across several complex and large datasets, including MNIST, fashion-MNIST, CIFAR10, CIFAR100, and STL-10** (Response Figs. 2 and 3).

- **We observed that incorporating random noise training at various network depths and on diverse datasets consistently resulted in higher final accuracy, often comparable to that achieved with backpropagation** (Response Figs.1 - 3).
- As the network depth increased, the gap in final accuracy between models with and without random noise training widened (Response Figs. 1, 3).
- Importantly, the beneficial impact of random noise training on final accuracy became significantly more pronounced as task complexity increased (Response Figs. 2, 3). For instance, while random noise training improved accuracy by 0.49% in MNIST, it increased by 3.04% in CIFAR10, 4.28% in CIFAR100, and 4.81% in STL-10. This gap widens further as the network becomes deeper.

Reviewer o5xh expressed concerns regarding the convergence of learning in our analyses. To address this, **we extended our training duration until convergence was confirmed, with validation accuracy showing no further increase** (patience: 10 epochs) (Response Figs. 1, 2). Additionally, in response to suggestions from reviewers o5xh and kKCK, **we have included numerical presentations of final accuracy to provide a clearer assessment of model performance.**

## Issue 2: Computational Benefits of Random Noise Training Considering Total Training Epochs (Pretraining + Data Training)

**We confirmed that random noise pretraining accelerates convergence and reduces computational resources, even when considering the total training duration (pretraining + data training)** (Response Fig. 4).

By varying the duration of noise training, we demonstrated that, despite the additional time required for noise training, the total training time remained significantly shorter than that of training with data alone in most conditions.

- We conducted subsequent data training on networks pretrained with random noise for 2, 5, 10, 20, and 50 epochs, measuring the epochs required for training to converge (validation accuracy no longer increasing, patience: 10 epochs). We maintained consistency in the number of samples used per epoch during both random noise training and subsequent data training, ensuring direct comparability of epoch times.
- We found that longer periods of random noise training resulted in shorter subsequent data training times to achieve convergence (Response Fig. 4a, Random + data vs. Data training only). As reported before, when comparing data training alone, the learning time consistently proved much shorter in networks pretrained with noise. Notably, **even when the time for noise training was added, the overall training duration for the noise-training algorithm remained substantially shorter than that for training with data alone in most conditions** (Response Fig. 4b).
- Based on this analysis, **we estimated an optimal duration for random noise pretraining that ensures the most efficient use of resources** (Response Fig. 4b, optimal). We also found that at this optimal duration, the improvement in accuracy remained significant.

These additional experimental results demonstrate that, considering the resources expended in random noise training, it represents a more efficient approach (reduced training time, overall computation) compared to training with data alone.

These results will also help address Reviewer o5xh's concern that random noise training may become too lengthy, by suggesting an effective duration for random noise training.

---

### Decision · Program_Chairs · 2024-09-25

**Decision:**

Accept (poster)

**Comment:**

This submission introduces a novel pretraining approach inspired by spontaneous neural activity. Using random noise with a feedback alignment algorithm, the authors demonstrate improved learning efficiency and generalization without relying on weight transport. The method aligns forward weights with synaptic feedback, leading to faster convergence and better performance across various tasks, including out-of-distribution scenarios. The additional experiments and clarifications provided during the rebuttal period further validate the findings. All reviewers voted for acceptance.